# Historical Review of Simultaneously Extracted Metal Copper Sediment Concentrations in Agricultural and Non-Agricultural Areas

**Lenwood W. Hall, Jr. \* and Ronald D. Anderson**

Wye Research and Education Center, Agricultural Experiment Station, College of Agriculture and Natural Resources, University of Maryland, Queenstown, MD 21658, USA; randers3@umd.edu
\* Correspondence: lwhall@umd.edu

**Abstract:** The objectives of this study were to: (1) summarize Simultaneously Extracted Metal (SEM) copper sediment concentrations based on a historic review from 54 study areas in 16 different countries with different land use activities and (2) compare SEM copper sediment concentrations from among the four land use categories: all study areas; non-agricultural study areas; agricultural study areas; and reference/control study areas. Based on over 1000 measurements, the mean SEM copper concentrations in non-agricultural area (26.0 μg/g) was higher than the mean concentrations for all sites (20.0 μg/g), agricultural sites (19.8 μg/g), and reference/control sites (3.87 μg/g). The 90th centile for non-agricultural sites (89.0 μg/g) was also higher than all sites (61.9 μg/g), agricultural sites (54.8 μg/g), and reference/control sites (17.1 μg/g). The maximum SEM copper concentration for the non-agricultural sites (902 μg/g) was approximately an order of magnitude higher than the maximum value for the agricultural sites (96.6 μg/g). The various potential sources of SEM copper may be responsible for the higher concentrations in non-agricultural areas, as the primary single anthropogenic source for copper in agricultural areas is likely copper use as a plant protection product (PPP). Future research efforts are recommended to expand the spatial and temporal scale of SEM copper sediment data, address actual SEM copper ecological risk to resident benthic communities with multiple stressor field studies, and compile a historical review of acid volatile sulfide (AVS) data used to determine bioavailable concentrations of SEM copper.

**Keywords:** simultaneously extract metal copper; land use; acid volatile sulfides; sediment

## 1. Introduction

Metals can enter aquatic ecosystems from various point and non-point sources due to both natural and anthropogenic causes [1]. Data on the total concentrations of metals in sediment have value but may not be sufficient to assess the ecological risk of metal polluted sediment because metals are present in different chemical forms that can impact bioavailability [2]. The toxicity of metals, such as copper, depends to a large extent on the type of binding forms [3]. For example, the bioavailability of metals such as copper in sediment is controlled by multiple factors such as physico-chemical (e.g., pH, redox potential, and particle size), geochemical (e.g., organic matter, metal dioxide, and sulfide), and biological (e.g., feeding behavior and uptake rates) [4]. A mechanistically determined method for predicting sediment toxicity due to metals in sediment that accounts for bioavailability through normalization to sediment sulfides is defined as simultaneously extracted metal to acid volatile sulfide (SEM/AVS) ratio. The process of normalizing sediment sulfides that can react with cationic metals to create insoluble metal sulfides reduces environmental availability and mobility of trace metals [5]. AVS is extracted from anaerobic sediment with cold 1 M Hydrochloric Acid (HCL) and divalent metals such as zinc, copper, cadmium, nickel, and lead released during this treatment are referred to as simultaneously extracted metals (SEM) [6,7].

Other investigators have reported that historical information on AVS concentrations in sediment used to determine SEM is scarce [8]. Although the 2007 paper published by these investigators is somewhat dated, in our view the need to summarize SEM copper (bioavailable potentially toxic form) based on AVS data still exists, as there is still no historical summary of SEM copper data available based on a variety of land use activities. However, there is one study where a brief review of SEM copper data based on just agricultural use has been conducted [9]. Anthropogenic sources for SEM copper in the aquatic environment include wastewater treatment plants, various industrial wastes, stormwater runoff, mining activities, antifouling boat paints, and agricultural use [10–12]. For example, copper-based compounds are used as fungicides to protect crops from important diseases in many areas of the world. The European Union considers copper-based compounds to be both persistent and toxic [13]. Copper fungicides have therefore been designated as a candidate for substitution (can be replaced by another pesticide). The European Food Safety Authority (EFSA) has outlined a framework for the environmental risk assessment (ERA) of transition metals, including copper, used as active substances in plant protection products (PPP). This new framework will be considered during the next renewal period for copper compounds [13]. Copper sediment monitoring data relevant for European agricultural streams has been identified as a regulatory need within this ERA framework.

The objectives of this study were to (1) summarize SEM copper sediment concentrations based on a historic review from 54 study areas in 16 different countries with different land use activities and (2) compare SEM copper sediment concentrations from among the four land use categories: all study areas; non-agricultural study areas; agricultural study areas; and reference/control study areas. Reasons why SEM copper concentrations may be different among the land use categories and ecological relevance issues related to SEM copper are discussed.

## 2. Materials and Methods

Google was primarily used to search for relevant SEM copper studies in the literature, and when the titles were located most of the reference materials were downloaded directly from journal websites via the University of Maryland Libraries system. In some cases, book chapters were accessed via the University Inter Library Loan system. Once the documents were obtained, they were scrutinized for relevance (proper type and quality of data) and the reference sections were examined for useful SEM copper articles that may not have been previously obtained.

References were searched for several primary variables that would eventually be inserted into the main manuscript historical summary as described below in Table 1. One of the variables needed for this table was the location of the sample sites so that the primary surrounding land use could be determined. Land use categories used were all sites, non-agriculture sites, agricultural sites, and reference/control sites. All sites were defined as all the sites in the data set. The non-agricultural site use category included urban, urban/forest, residential, copper mining in area, historic metal contamination, and harbor/ports (i.e., antifouling paints). The agricultural land use site category were sites dominated by agricultural use. The reference/control site category was based on using the terminology used by the authors within the various papers or our judgement of location based on Google Earth.

**Table 1.** Summary of historical sediment SEM copper data (µg/g dry wt) based on all land use activities.

| Location | Water Body/Primary Surrounding Land Use | Depositional Areas Targeted? | # of Sites Sampled and Frequency | SEM Cu (Min-Max, Mean) | Reference |
|---|---|---|---|---|---|
| SE Coast of Brazil | Sergipe River Estuary and 2 tributaries/Primarily urban | Not reported | 3 sites, 1 core each, 21 total samples from various core depths | Sal River: (22.6–40.0, 32.6) Sergipe River: (15.6–34.2, 25.7) Poxim River: (10.5–23.9, 16.3) | [14] |
| Washington State, USA | Steilacoom Lake/Urban-Residential (heavy algaecide use) | Not reported | 10 sites with measured concentrations sampled once | Steilacoom Lake: (38.1–194, 108) | [7] |
| Western Montana, USA | Upper Clark Fork River and Milltown Reservoir/Primarily forested but with some inputs from agriculture or Cu mining upstream | Not reported | 3 sections, 7 total sites sampled once from composite samples (1 of 7 sites agr) | RC (reference): (1.91–1.91, 1.91) [a] Milltown Reservoir: (35.0–902, 265) Upper Clark Fork: (655–655, 655) Upper Clark Fork Agriculture (CF4): (36.2–36.2, 36.2) | [15] |
| Sweden and Denmark | SW Sweden/Agriculture [b] E Denmark/Agriculture | Yes (sand grain size and smaller) | Sweden: 3 sites, 1 composited sample each site, sampled once [c] Denmark: 6 sites | Sweden: (2.29–8.90, 4.85) Denmark: (1.84–110.2, 5.28) | [8] |
| England, Finland, Belgium, France, Germany, and Italy | S England and Wales/Agriculture S Finland/Agriculture S Belgium/Agriculture N France/Agriculture W and S Germany/Agriculture N Italy/Agriculture | Yes | England/Wales: 16 sites Finland: 5 sites Belgium: 6 sites France: 12 sites Germany: 9 sites Italy: 2 sites | England/Wales: (7.12–25.9, 15.4) Finland: (1.21–41.5, 13.0) Belgium: (3.50–22.7, 11.2) France: (0.064–20.8, 8.23) Germany: (1.97–76.1, 15.7) Italy: (3.05–3.56, 3.30) | [8] |
| Guadalete Estuary and SW Spain (tidal sites) | Site G1/Harbor/Port Sites G2–G3/Agriculture Sites S1–S7/Mouths of agriculture drains | Yes (most samples <63 um) | 10 sites with 3 replicates/site, sampled twice (Aug 2002 and Mar 2003) | Site G1: (4.4–170, 46.4) Sites G2–G3: (10.8–16.5, 14.0) Sites S1–S7: (5.7–21.0, 14.4) | [3] |
| Shenzhen Bay, SE China | Mangroves influenced by the Fengtanghe and the Shenzhenhe Rivers/Primarily urban | Not reported | 16 sites sampled once with 3 replicates per site | Sites 1–16 (means): (0–21.6, 9.47) Max value, (34.1, n = 48) [d] | [16] |
| N Belgium, E of Antwerp | Lowland riverine sediments known to have historic metals contamination/Primarily agriculture-forest [e] | Not reported | 17 sample sites (3 replicates each) sampled once | Sites 1–17 (means): (2.29–40.6, 14.6) | [17] |
| N Belgium, E of Antwerp | Lowland riverine sediments known to have historic metals contamination/Primarily agriculture-forest [e] | Not reported | 28 sample sites (3 replicates each) sampled once | Sites 1–28 (means): (6.15–712, 44.3) [d] | [18] |
| Flanders Region, N Belgium [f] | Navigable and non-navigable watercourses/Land use unknown | Yes (sandy to silty) | 200 sites sampled once | Sites 1–200: (0.0–62.3, 8.65) | [19] |

**Table 1.** *Cont.*

| Location | Water Body/Primary Surrounding Land Use | Depositional Areas Targeted? | # of Sites Sampled and Frequency | SEM Cu (Min-Max, Mean) | Reference |
|---|---|---|---|---|---|
| Antioch, California State | Lower Kirker Creek/Urban Upper Kirker Creek/Agriculture [f] | Yes | 14 sites with composite samples collected once for 2 years | 12 Urban Sites: (0.191–12.6, 3.05) 2 Agriculture Sites: (0.191–12.4, 4.21) | [20] |
| Sacramento, California State | Arcade Creek/Urban | Yes | 11 sites with composite samples collected once/year for 3 years | 11 Urban Sites [g]: (0.95–11.1, 4.71) | [21] |
| Salinas, California State | Alisal, Gabilon and Natividad Creeks/Urban with some agriculture | Yes | 13 sites with composite samples collected once/year for 3 years | 11 Urban Sites [g]: (3.62–20.6, 8.49) 2 Agriculture Sites: (2.67–8.26, 5.46) | [21] |
| N Illinois State | Big Bureau Creek/Agriculture | Yes | 12 sites with composite samples collected once/year for 3 years | Sites 1–12: (2.54–7.63, 4.63) | [22] |
| Santa Maria, California State | Santa Maria River, Osco Flaco Creek, Orcutt Creek and unnamed drainages/Agriculture | Yes | 12 sites with composite samples collected once/year for 3 years | Sites 1–12 [g]: (7.63–20.3, 11.5) | [23] |
| Pleasant Grove, California State | Upper Pleasant Grove Creek/Urban Lower Pleasant Grove Creek/Agriculture | Yes | 21 sites with composite samples collected once/year for 10 years | 18 Urban Sites: (0.508–252, 21.5) 3 Agriculture Sites: (0.191–21.9, 6.94) | [24] |
| Rio Vista California State | Cache Slough/Agriculture | Yes | 12 sites with composite samples sampled twice/year for 3 years | 12 sites: (8.9–59.1, 23.8) | [25] |
| N Belgium | Nete/Scheldt River Basins/sediments known to have historic metals contamination/Unknown | Not reported | 1 control sediment site (Alava, Spain) 3 sites sampled once | Control site: (1.91–1.91, 1.91) 3 Sites: (0.64–146, 56.3) | [26] |
| SE Coast of Brazil | Three rivers of the Santos-Cubatao estuarine system/Urban-Forest | Not reported | 3 sites sampled once or twice (winter and/or summer) | 3 sites: (<0.127–52.1, 8.65) | [27] |
| Washington State Desert | Hanford Reach (Columbia River)/Desert Priest Rapids Dam (Columbia R)/Agriculture McNary Dam (Columbia River)/Agriculture Ice Harbor Dam (Snake River)/Agriculture | Not reported | 4 sites sampled 2–3 times over 3 years 6 sites sampled 2–3 times over 3 years 6 sites sampled 2–3 times over 3 years 3 sites sampled 2 times over 2 years | Hanford Reach Site: (5.27–12.6, 8.63) Priest Rapids Dam Sites: (4.38–30.5, 17.7) McNary Dam Sites: (6.80–20.9, 15.9) Ice Harbor Dam: (4.64–15.7, 12.3) | [28] |
| Ravenna, NE Italy (tidal sites) | Pialassa Piomboni (coastal lagoon)/Primary freshwater input from agriculture | Not reported | 50 sites sampled once | Pialassa Piomboni: (0.318–89.0, 6.35) | [29] |
| SE Netherlands | Beekloop (headwater stream)/Agriculture [h] | Not reported | 4 sites sampled once with 3 replicates per site | Sites L1–4: (19.1–76.3, 53.2) | [30] |
| N Serbia | Various rivers, canals, streams/Agriculture Various rivers, canals, streams/Urban | Yes | 9 urban sites sampled twice in one year 3 urban sites sampled twice in one year | Agriculture: (6.35–96.6, 45.4) [i] Urban: (12.7–23.5, 17.6) | [2] |
| SE Coast of Australia | Boronia Park, Lane Cove Estuary/Urban | Not reported | One reference site sampled once | Reference site: (25.0–25.0, 25.0) | [31] |
| SW Coast of India | Vembanad Lake System Estuary/Urban | Not reported | 12 sites sampled over 3 years during the pre, post and monsoon periods | Sites 1–12: (0.635–35.6, 9.53) | [32] |

**Table 1.** *Cont.*

| Location | Water Body/Primary Surrounding Land Use | Depositional Areas Targeted? | # of Sites Sampled and Frequency | SEM Cu (Min-Max, Mean) | Reference |
|---|---|---|---|---|---|
| SE Coast of Australia | Cooks River Estuary/Urban | Not reported | Control sediment sampled once | 1 control site: (7.43–7.43, 7.43) | [33] |
| SW Netherlands | Meuse/Rhine River Delta /Agriculture [j] | Yes | 4 sites sampled twice in Nov 1995 and once in Jun 1996 | Sites 1–4: (19.1–76.3, 56.7) | [34] |
| N Netherlands | Lake Ketel/Agriculture (some urban upstream) | Yes (most sediment <63 um) | 4 sites (10 reps per site) sampled once | Sites A–D: (13.3–58.5, 35.3) | [35] |
| Netherlands/Belgium | Various Coastal Sites (11–20 km offshore) Various Urban Sites Various Agriculture Sites (some urban upstream) | Not reported | 8 sites sampled once 10 sites sampled once 3 sites sampled once | Coastal sites: (0.635–2.52, 1.27) Urban sites: (1.27–49.6, 17.1) Agriculture sites: (10.8–71.8, 34.5) | [36] |
| SE Coast of China | Maluan Bay/Urban-Industrial | Yes | 8 sites sampled once with 3 replicates each | 8 sites, ML1-ML8: (20.0–147, 68.9) [d] | [37] |
| NE Coast of China | Meiliang Bay and Wuli Lake/Urban | Not reported | 7 sites sampled once | 7 sites: (9.53–43.8, 23.3) | [38] |

[a] Not reported whether sample concentrations were dry weight measurements. [b] All sample sites in this study reported to be wadable streams with no evidence of nearby point source chemical or organic inputs. The map scale was too small to locate specific sites on Google Earth but all appeared to be located in rural areas where agriculture followed by forest was the dominant land use. In addition, sites that appeared to be high quality (as indicated by biological indicators) were targeted for sampling according to the authors. [c] All sites in this study had one composite sample, sampled one time. [d] These data were extracted from a data plot (not a data table). [e] All land use delineations reported for this study are gross estimates based on the small-scale map locations provided in document. [f] These sites surrounded by grazing land. [g] All data reported were 3 year mean values by site. [h] Authors report soils in region may be contaminated with heavy metals from historic ore smelting activities and manure from intensive livestock farming. [i] Maximum value from a site located at a ship lock in a canal. [j] Authors report that both the aquatic sediments and soils in the delta are contaminated by heavy metals.

Determining the surrounding land for the various study areas was often challenging. The references were searched for coordinates and/or maps of sample sites, as well as any descriptions in the text; alternatively, direct contact with the authors of the papers was sought to explain the site locations or surrounding land use. Google Earth was used to evaluate the land use around the sample sites if coordinates were given or if there were detailed maps showing the site locations. Otherwise, descriptions from the text and best professional judgement were used to locate sites using Google searches. In some cases, the only site information available was a very small-scale map, which made it challenging to determine surrounding land use. In some cases, obviously wrong or unusable coordinates were given, and in most cases no coordinates were provided.

Copies of the relevant raw data were transferred to Excel spreadsheets for later analyses (see Data Availability Statement). In some cases, copies of figures with SEM copper data were extracted from the document so that a close approximation of the data in the figure could be transcribed to spreadsheets. This involved creating a larger copy of the figure and the use of a micrometer to make fine measurements to convert into SEM copper concentrations. All SEM copper concentrations were converted into μg/g values to have consistent units for all the references.

The information for each reference with relevant SEM copper sediment data was organized in Table 1 with the following categories: (1) location; (2) waterbody type and primary surrounding land use; (3) if deposition areas were targeted; (4) number of sites sampled and frequency of sampling; (5) SEM copper concentrations including minimum, maximum, and mean; and (6) reference. The SEM copper data were placed in the following four categories for the analysis described below: all study areas, non-agricultural study areas, agriculture study areas, and reference/control study areas. The approximate locations of the various study areas are presented in Figures 1–3.

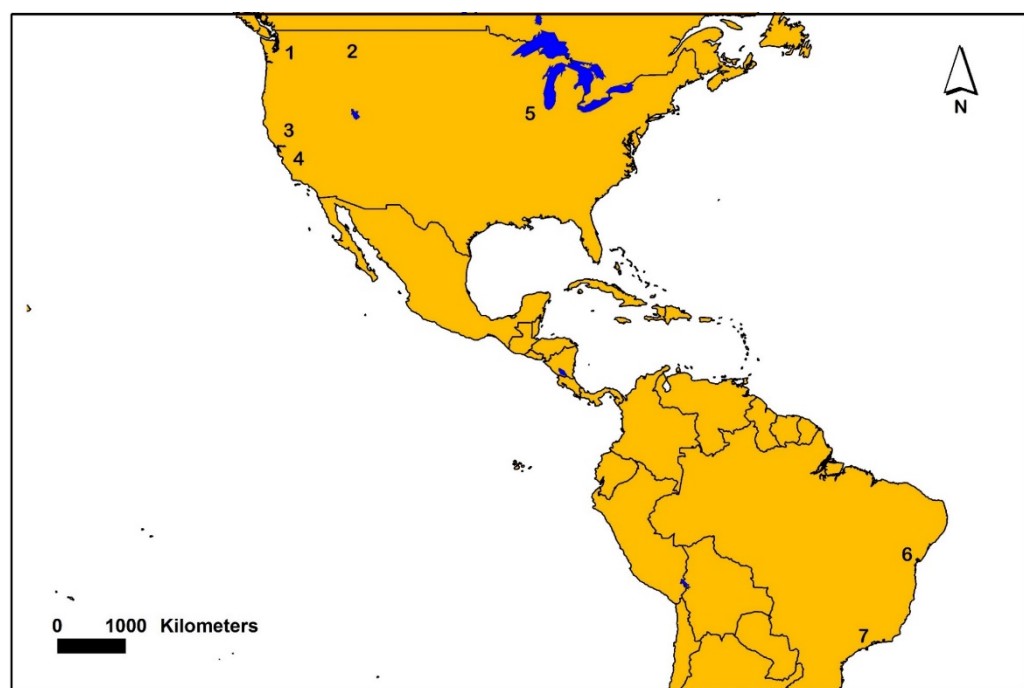

**Figure 1.** Map showing generalized locations where SEM Cu was sampled from various studies in North and South America. Number symbols on the map are associated with individual or multiple studies and references. The following numbers and associated references are: 1 (7, 28), 2 (15), 3 (20,21,24,25), 4 (21,23), 5 (22), 6 (14) and 7 (27).

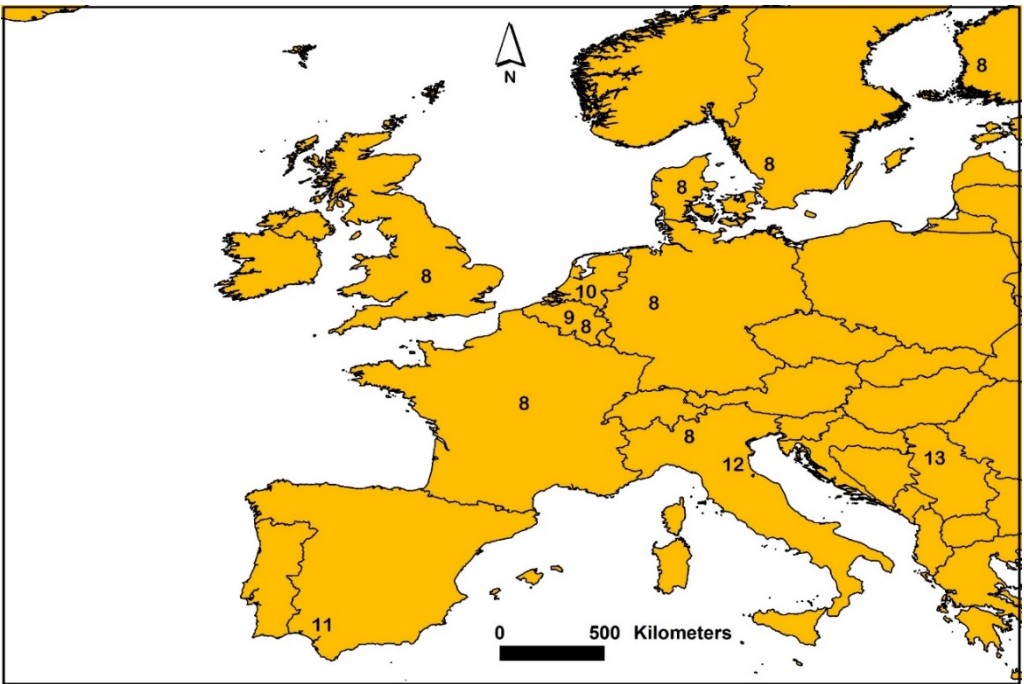

**Figure 2.** Map showing the generalized locations where SEM Cu was sampled from various studies in Europe. Number symbols on the map are associated with individual or multiple studies and references. The following numbers and associated references are: 8 (8), 9 (17,18,19,26), 10 (30,34,35,36), 11 (3), 12 (29) and 13 (2).

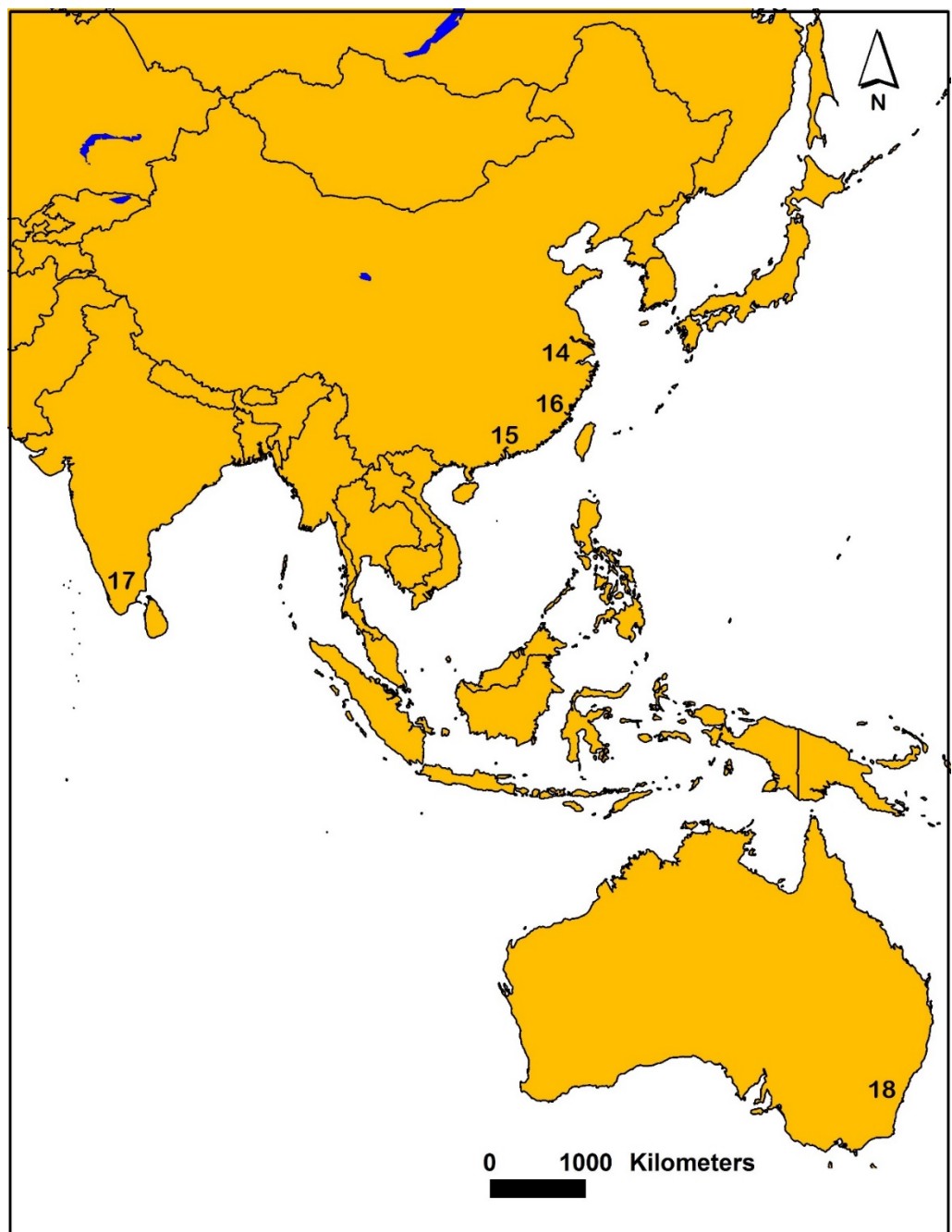

**Figure 3.** Map showing the generalized locations where SEM Cu was sampled from various studies in India, China, and Australia. Number symbols on the map are associated with individual or multiple studies and references. The following numbers and associated references are: 14 (38), 15 (16), 16 (37), 17 (32) and 18 (31,33).

The statistical analysis of raw data using SigmaPlot (SYSTAT, www.systat.com (accessed on 2 February 2022)) was used to calculate the SEM copper mean (with standard deviation), range, and 90th centile for each of four different categories of data: all study areas; non-agricultural study areas; agricultural study areas; and reference/control study areas. The 90th centile means that 90% of the values were below this centile. Using SigmaPlot, the SEM copper concentrations were ranked from low to high and a regression plot was produced with a probability scale on the *y*-axis and a log scale on the *x*-axis (SEM copper concentration). The a and b factors of the regression equation were used in the following equation to calculate the 90th centile: $10^{((\text{probit}\% - (a+5))/b)}$ where: probit % = the

probit transformed percentage (i.e., if a 90th centile is desired then the probit transformed percentage equal to 90% was used).

## 3. Results

The results from the literature review of SEM copper data in Table 1 showed that data were available from 54 different study areas from 16 different countries. Countries and number of referenced areas per country were as follows: Brazil (2), United States (9), Sweden (1), Denmark (1), England/Wales (1), Finland (1), Belgium (5), France (1), Germany (1), Italy (2), Spain (1), China (3), Netherlands (3), Serbia (1), Australia (2), and India (1). In some cases, there were multiple study areas contained in a single reference.

The U. S. had the highest number of SEM copper references (nine) while nine different countries were represented by only one reference. Therefore, the database was somewhat bias for U. S. sediment, particularly for California (six study areas). Depositional areas were targeted for approximately half of the references, and for the other areas the type of sediment sampled was unknown. Depositional areas are important to consider because copper tends to accumulate in these fine grain areas.

Statistical analysis of SEM copper data by study category (all sites, non-agricultural sites, agricultural sites, and control/reference sites) from over 1000 measurements is presented in Table 2. The mean SEM copper concentration for the reference/control sites (3.87 µg/g) was substantially lower than the other three categories. The mean SEM copper concentrations for the non-agricultural sites (26 µg/g) was approximately 14% higher than the mean value for the agricultural sites (19.8 µg/g). However, the mean for agricultural sites was similar to the mean for all sites (20.0 µg/g). The maximum value for the non-agricultural sites (902 µg/g) was approximately an order of magnitude higher than the maximum value for the agricultural sites (96.6 µg/g). This maximum value for the agricultural sites was also substantially higher than the maximum value for the reference/control sites (25 µg/g).

**Table 2.** Descriptive statistics and centile calculation for four different categories of studies with SEM copper field data results. The European Copper Institute (2008) [19] study (Table 1) with unknown land use was left in the database for statistical analysis of the All Studies category but not included in the database for the non-agricultural analysis.

| Study Category | N, Mean, SD | Min, Max Values | 90th Centiles |
|---|---|---|---|
| All Studies | 1021, 20.0, 47.5 | 0.0, 902 | 61.9 |
| Non-Agricultural | 425, 26.0, 70.8 | 0.0, 902 | 89.0 |
| Agriculture | 384, 19.8, 16.8 | 0.0635, 96.6 | 54.8 |
| Reference/Control | 12, 3.87, 6.90 | 0.635, 25.0 | 17.1 |

In order to provide an analysis of SEM copper data distribution by category type, 90th centile calculations were conducted as presented in Table 2 and Figures 4–7. The 90th centile for the all sites category 61.9 µg/g (Figure 4) and agricultural sites 54.8 µg/g (Figure 6) is similar. For the non-agricultural sites in Figure 5, the 90th centile of 89 µg/g was higher than the 90th centile for the agricultural sites (54.8 µg/g) in Figure 6. Particularly noteworthy is the upper tail of the distribution in Figure 5, where the number of sites with SEM copper concentrations above 100 µg/g for the non-agricultural sites was substantial, while all SEM copper concentrations were less than 100 µg/g for the agricultural sites (Figure 6). These data clearly demonstrate that SEM copper concentrations were higher in non-agricultural areas compared with agricultural areas. As expected, the SEM copper 90th centile of 17.1 µg/g for reference control/sites was much lower than the other three categories (Figure 7). All of these reference/control site values were less than 8 µg/g except one.

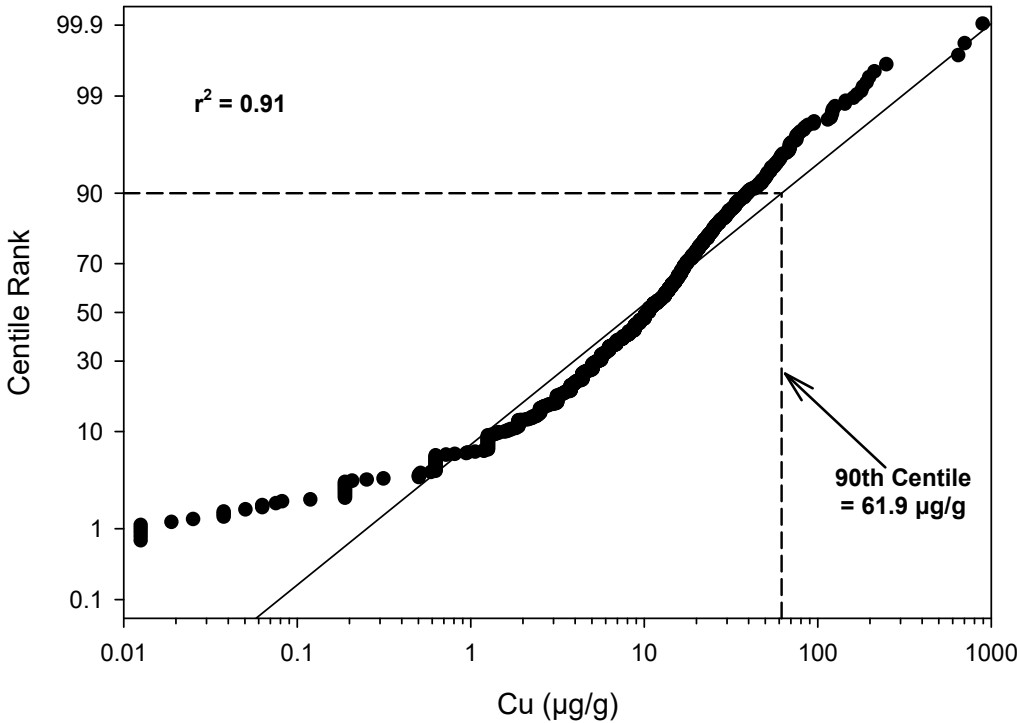

**Figure 4.** Regression of All SEM copper field study raw data against a probability scale showing the 90th centile rank.

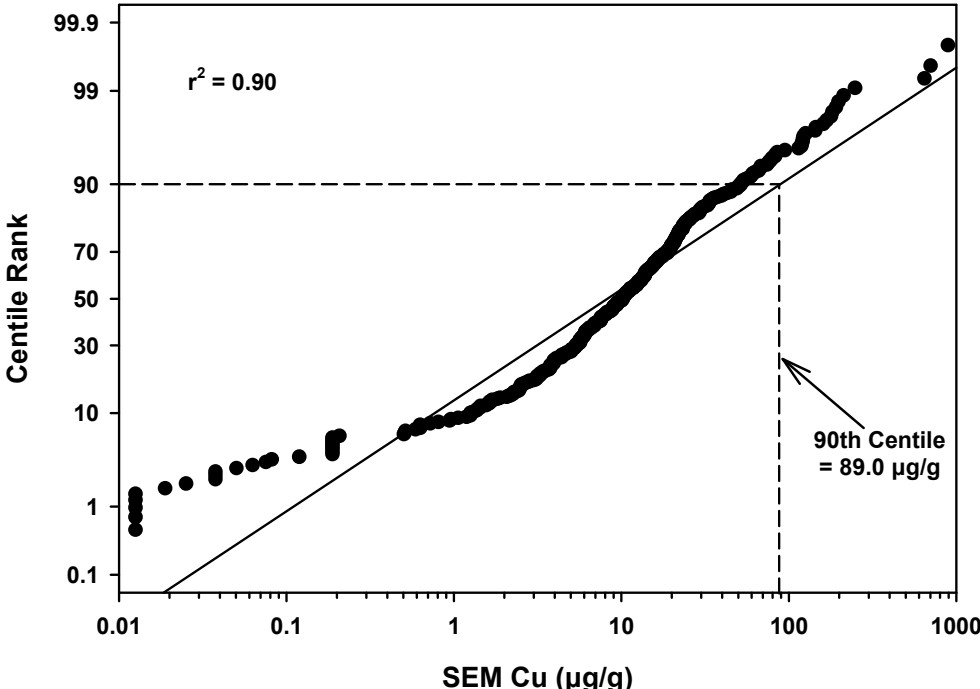

**Figure 5.** Regression of Non-Agricultural SEM copper field study raw data against a probability scale showing the 90th centile rank.

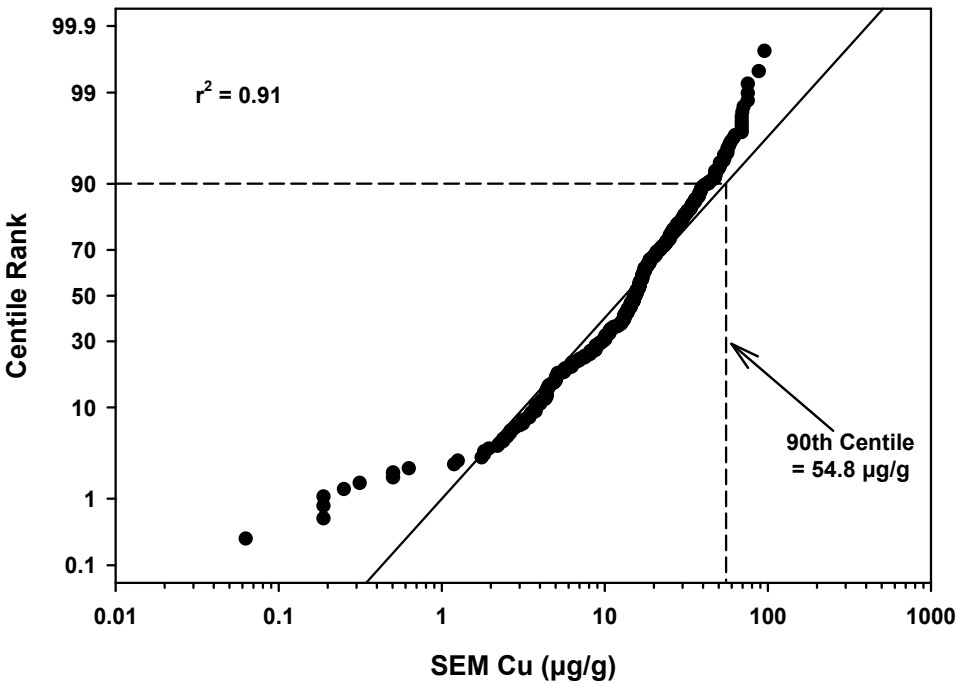

**Figure 6.** Regression of Agricultural SEM copper field study raw data against a probability scale showing the 90th centile rank.

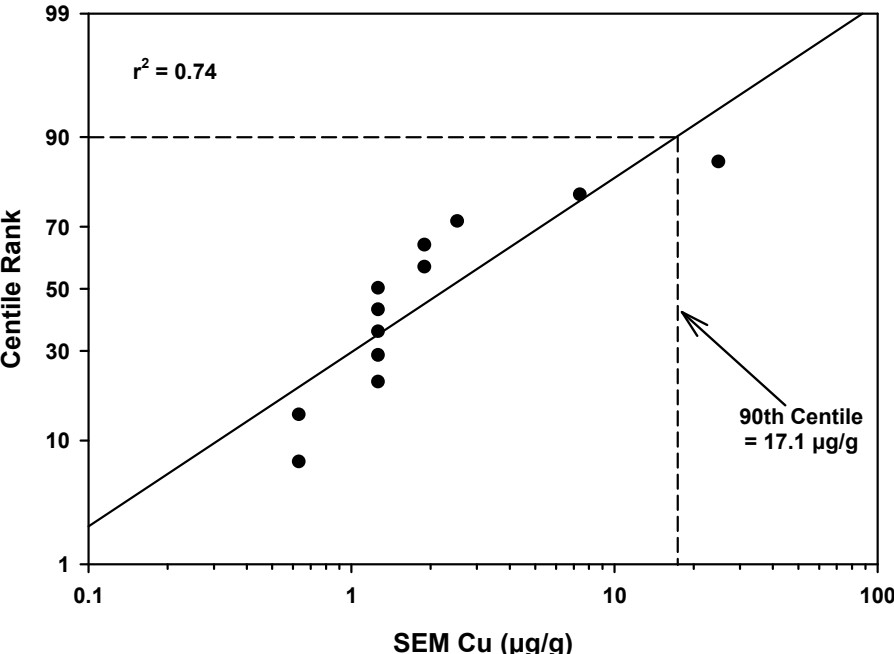

**Figure 7.** Regression of Control/Reference SEM copper field study raw data against a probability scale showing the 90th centile rank.

## 4. Discussion

Results from the analysis presented above demonstrated that SEM copper concentrations based on both mean values and 90th centiles were higher in non-agricultural areas when compared with all sites, agricultural areas, and reference/control areas. There were numerous potential sources of SEM copper from non-agricultural areas such as urban activity with imperious surfaces (e.g., brake pads from vehicles), wastewater treatment plants, various industrial wastes, residential activity, copper mining in area, historic metal

contamination, and harbor/ports (e.g., antifouling paints). These various potential sources of SEM copper may be responsible for the higher concentrations in non-agricultural areas as the primary anthropogenic source for copper in agricultural areas is copper use as a PPP. The regulatory implications of these results are important when considering the future use of copper as a PPP (fungicide). From a land use prospective, copper risk is clearly much higher in non-agricultural areas and copper concentrations in both non-agricultural and agricultural areas exceed concentrations in reference/control areas. If the protection goal for copper is to reduce SEM copper concentrations to a reference/control mean of 3.87 µg/g and a 90th centile of 17.1 µg/g, then significant changes in copper use would be needed based on all activities that may contribute to copper concentrations in the aquatic environment. However, before considering such a drastic step, it would be wise to determine if copper is actually impacting resident benthic communities (bioassessments) based on multiple year field studies where multiple stressors are evaluated along with SEM copper in both agricultural areas and non-agricultural areas (urban/residential areas). Results from this field-based bioassessment multiple stressor approach would provide an "impact observed response" to SEM copper in contrast to the use of laboratory toxicity data used to determine regulatory standards that provide an "impacted predicted response" [39]. The limited available data that can be used to address this ecological relevance question in both agricultural and non-agricultural areas are presented below.

A multiple stressor–multiple year bioassessment study in an agricultural waterbody in California (Cache Slough) where copper is used as a PPP was conducted to determine the relationship between various benthic metrics and sediment characteristics, metals (bulk metals including copper and SEM/AVS including SEM copper), and pyrethroids [25]. The relationship of 11 benthic metrics representing richness, composition, tolerance/intolerance, and trophic measures to 28 different stressors was evaluated (including bulk copper and SEM copper) over a three year period. The SEM copper concentrations in this study area ranged from 8.9 to 59.1 µg/g [9]. The results from this study showed that resident benthic communities in an area dominated by agriculture with reported use of copper as a PPP appear to be more closely associated with sediment characteristics and some metals but not copper or pyrethroids. Therefore, the field evidence from this study where copper is clearly used as a PPP and measured in sediment is that neither bulk copper nor SEM copper is impacting resident benthic communities.

A similar study design was also used in an urban/residential area based on sampling 21 sites for 10 years in Pleasant Grove in Roseville, California [24]. The goals of this study were to determine the relationship of various benthic metrics to physical habitat metrics, pyrethroids, metals (including bulk copper and SEM copper), and sediment characteristics. SEM copper concentrations ranged from 0.058 to 252 µg/g in this study area. The results from this field study showed that certain physical habitat metrics indicative of stream flow, hydrology, habitat diversity, and substrate quality overshadowed any apparent effects of pyrethoids or metals (including bulk copper and SEM copper) on shaping benthic communities when all variables were considered in multivariate analysis. Therefore, this is another line of evidence from a non-agricultural use area where SEM copper was measured in sediment but not found to be an important stressor to resident benthic communities.

The summary of historical SEM copper data presented in this paper can be used to identify data gaps that should be addressed. Despite available SEM copper data from 16 countries, the spatial and temporal scale of the data should be improved. The number of SEM copper studies exceeding a single year is limited based on this historical review. In addition, the number of sites sampled in the various studies was also minimal. Additional field studies where benthic communities are exposed to SEM copper due to copper use along with other chemical and non-chemical stressors are also needed to provide "real world evidence" that SEM copper is actually impairing resident communities. The current database for this type of research is primarily in California, therefore the spatial scale needs to be expanded. The scale of this proposed fieldwork should be adequate to address multicollinearity issues. Another research gap that needs to be addressed is a current

summary of all historical AVS data from various regions of the world. Other investigators recognized this research need approximately 15 years and ago, and the recommendation from these authors still exists [8]. It is important to know the typical AVS concentrations from various freshwater ecosystems due to the strong relationships between AVS and bioavailable concentrations of metals such as copper, as well as other metals.

## 5. Conclusions

SEM copper sediment concentrations were higher in non-agricultural areas when compared to agricultural areas based on an analysis of over 1000 measurements from 16 different countries. The various activities resulting in multiple sources of SEM copper from non-agricultural areas primarily compared with a single anthropogenic source in agricultural areas (PPP) was the likely reason for the higher concentrations. SEM copper concentrations from both non-agricultural areas and agricultural areas were substantially higher than concentrations reported from reference/control areas. A significant change in copper use would be needed in both non-agricultural and agricultural areas to lower SEM copper concentrations closer to the reference concentrations. However, such an action should not be considered unless bioassessment multiple stressor field studies (including SEM copper) demonstrate that SEM copper is actually impacting resident benthic communities in both non-agricultural and agricultural areas. To date, available limited data using this approach in both agricultural and non-agricultural areas demonstrate that ambient concentrations of SEM copper are not impacting resident benthic communities. However, these ecologically relevant field studies are scarce and need to be expanded.

**Supplementary Materials:** The following supporting information can be downloaded at: https://www.mdpi.com/article/10.3390/agriculture12050711/s1, Hall & Anderson (2022) Supplemental SEM Cu Data.

**Author Contributions:** Conceptualizations and writing, L.W.H.J.; historical data searches and development of tables and figures, R.D.A. All authors have read and agreed to the published version of the manuscript.

**Funding:** Albaugh is acknowledged for supporting the collection and analysis of data.

**Institutional Review Board Statement:** Not applicable.

**Informed Consent Statement:** Not applicable.

**Data Availability Statement:** The data are available as a separate file (Supplementary Materials).

**Acknowledgments:** Robert Morris is acknowledged for his constructive comments on the draft manuscript.

**Conflicts of Interest:** The authors declare no conflict of interest.

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
