# Peer review of "Historical Review of Simultaneously Extracted Metal Copper Sediment Concentrations in Agricultural and Non-Agricultural Areas"

_agriculture, doi:10.3390/agriculture12050711_

Round 1

Reviewer 1 Report

Dear Authors,

Please refer to these comment and suggestions.

Title:

The title is acceptable as Review Paper. However, agriculture and non-agriculture area as a key point of the study is suggested to be put in the title.

Abstract:

Unclear of the Problem statement, methodological approach, and conclusion. For the objective please refer to the objective in the Introduction.

Introduction:

  • The elaborate clear problem statement and the significance of this review study
  • However, the lack of explanation for why choose these 16 countries (54 study areas) are to be studied/compared.  An unclear gap in research.

Material and Methods

  1. Clear Research area.
  2. The review method is acceptable.

 Results:

  • Result is acceptable
  • However, it is still a lack of discussion, especially relating to agriculture and non-agriculture area..

 Conclusion

  • Acceptable conclusion, however, need an impact and realistic recommendations based on this review study.

Thank you.

Author Response

My response to each comment by Reviewer 1 is as follows:

The title is acceptable in Review Paper. However, agriculture and non-agriculture areas are as a key point of the study is suggested to be put in title.

The title was changed as suggested by the reviewer.

Abstract:

Unclear of the Problem statement, methodological approach, and conclusion. For the objective please refer to the objective in the Introduction.

The objective used in the Introduction was used in the Abstract as suggested by the reviewer.

Introduction: 

The elaborate clear problem statement and the significance of this review study.

I am not clear on this comment. However the significance of this work and why it is unique is clearly stated in the Introduction.

However, the lack of explanation of why choose 16 countries (54 study areas) are to be compared. An unclear gap in research.

The 16 countries and 54 study areas were used because these were the only locations where SEM copper data were available. 

Materials and Methods

  1. Clear Research Area
  2. The review method is acceptable

No response needed

Results: 

Result is acceptable - no response needed

However, it is still a lack of discussion, especially relating to agriculture and non-agriculture area.

Agricultural and non-agricultural sources of SEM copper are discussed in the first paragraph of the discussion section

Conclusion;

Acceptable conclusion, however, need an impact and realistic recommendation based on this review.

See line 287 for the recommendation for future work as suggested by the reviewer

Reviewer 2 Report

This manuscript titled in “Historical Review of Simultaneously Extracted Metal Copper Sediment Concentrations” reports a summary of simultaneously extracted metal (SEM) copper sediment concentrations based on a historic review from 54 study areas in 16 different countries with different land use activities and a comparison of SEM copper sediment concentrations from four land use categories. This manuscript discusses reasons why SEM copper concentrations may be different among the land use categories and ecological relevance issues related to SEM copper. This work is interesting and reasonably well executed. The authors should consider the following points in any revision as follows:

  1. The authors should explain the meaning of the four land use categories: all study areas; non-agricultural areas; agricultural study areas and reference/control areas (especially the all study areas and reference/control areas).
  2. The authors should explain the meaning of 90th centile in order that the readers can understand what it’s value means.
  3. Generally speaking, legends should be at the bottom of the figures.
  4. On Page 10, Line 176, it’s better to list and label the equation to calculate the 90th centile on a single line.
  5. The references were published in different years, and the time range is over twenty years. Also, the targeted countries and areas had different developmental levels. So, are the results reasonable here?
  6. Although now bioassessment multiple stressor field studies don’t demonstrate that SEM copper is actually impacting resident benthic communities in both non-agricultural and agricultural areas, we are not sure whether SEM copper is harmful to our environment. So, we still should consider the action to lower SEM copper concentrations.
  7. How to predict the bioavailability toxicity effect concentrations of SEM copper?
  8. Are the bioavailability toxicity effect concentrations of SEM copper in different areas inconsistent? What is the standard?
  9. Whether there is a significant relationship between pH and concentrations of SEM copper in the non-agricultural and agricultural study areas?
  10. What are the main sources of concentrations of SEM copper in the non-agricultural study areas?
  11. The areas represented by number symbols in Figure 1-3 can be further divided clearly so that they can be more intuitively clear. And in the figures, please pay attention to format of “th”.
  12. The authors should check the manuscript carefully to unify the reference format. For example, Reference 19, 21, 22, 24.

Author Response

My response to Reviewer 2 comments are as follows:

  1. The authors should explain the meaning of the four land use categories: All study areas; non-agricultural study areas; agricultural study areas; and reference/control  areas (especially the all study areas and reference/control areas)

All  of these site categories are explained in detail in the second paragraph of the Materials and Methods.

2, The authors should explain the meaning of the 90th centile in order that the reader can understand what its value means.

The is a good comment and definition of 90th centile was included in the text on line 172

3. Generally speaking the legends should be at the bottom of the figures.

The reviewer is correct and this is a change that the publisher can make when the proofs are prepared.

4. On page 10, line 176 it's better to list and label the equation to calculate the 90th centile on a single line. 

I am not clear on this comment as this equation is on a single line. I don't know any other way to present this equation.

5. The references were published in different years and the time range is over twenty years. Also the targeted countries and areas had different development levels. So are the results reasonable here.

The goal was to find all available SEM copper data available regardless of the year, geographic area, and level of development. Therefore, the results from our study are reasonable and address the goals of the study,

6. Although now (I think new was intended) bioassessment multiple stressor field studies don't demonstrate that SEM copper is actually impacting resident benthic communities in both non-agricultural and agricultural areas, we not sure whether SEM copper is harmful to the environment. So we still should consider action to lower SEM copper concentrations. 

This is more of regulatory/policy comment concerning lower SEM copper concentrations that is really not the focus of this paper. I only referenced two bioassessment multiple stressor studies where SEM copper was not found to impact benthic communities. However, in the last paragraph of the Discussion, I  make various recommendations (including the need for more bioassessment multiple stressor field studies where SEM copper is one of the stressors evaluated). If  SEM copper reduction is in fact needed the potential risk of copper replacements needed to be thoroughly assessed.

7. How to predict the bioavailability toxicity effect concentrations of SEM copper?

The bioavailability issue of SEM copper is addressed in detail in the Introduction.

9. Whether there is a significant relationship between pH and concentrations of SEM copper in the non-agricultural and agricultural study areas?

This is not a goal of the study (addressing the relationship of pH) and for some of the areas pH is not reported.

10. What are the main sources of concentrations of SEM copper in the non-agricultural areas?

Sources of SEM copper are in non-agricultural areas are described in detail on lines 80 -82 (second paragraph of the Materials and Methods)

11. The areas represented by number symbols in Figure 1-3 can be further divided clearly so that they can be more intuitively clear. And in figures please pay attention to format of "th".

These figures are the best we can do using Google Earth and in some cases no actual coordinates but only road crossing and other land marks are available. The intention of these figures is to only provide approximate site locations and show where SEM copper data were available on a global spatial scale.

I don' know understand the comment concerning the format of the "th".

12. The authors should check the manuscript format carefully to unify the reference format. For example 19, 21,22,24.

The reference format used followed the style guide for Agriculture. The Editorial Group certainly let me know if the format used is incorrect or more information is needed.

Reviewer 3 Report

The manuscript discusses the historical review of simultaneously extracted metal copper sediment concentrations. This is an important topic of research.
The following are my comments:
1) State the novelty of this study clearly.
2) Add a flow chart to demonstrate the steps followed in this study.

3) Is the number of data points enough for such a type of study? 
4) The recent literature needs to be cited
a) https://doi.org/10.1016/j.jconhyd.2018.11.002
b) https://doi.org/10.1038/s41598-020-68198-6

Author Response

My response to Reviewer 3 comments are as follows:

  1. State the novelty of the study clearly.

Paragraph 2 in the Introduction clearly states why this study is unique and contributes to the literature. The main reason for doing this study was to summarize historical SEM copper from all land use activities because this type of historical summary has not been done and it will provide useful information on the spatial scale of SEM copper. 

2. Add a flow chart to demonstrate the steps followed in this study.

I don' think a flow chart is needed because all the steps are clearly presented in the Methods and Materials (Use of Google and other tools to search for relevant data, organization of SEM copper data in Table 1, statistical analysis etc)

3. Is the number of data points enough for such a type of study?

The short answer is generally  yes as we have over 1000 SEM copper data points. The data distribution figures with 90th centiles in Figures 4, 5, and 6 are certainly robust. The number of data points for Figure 7 (control/reference areas) could certainly be improved but unfortunately this is all the data. I don't think this distracts from the key results from this study.

4. The recent literature needs to be cited:

(a) https://doi.org/10.1016/j.jconhyd.2018.11.002

(b) https://doi.org/10.1038/s41598-020-68198-6   

I check both of these papers and they do not contain SEM copper so they are not useful of this review paper.

Round 2

Reviewer 1 Report

Dear Sir,

Please refer to these comments.

Title:

The title is acceptable as Review Paper.

Abstract:

Unclear of the Problem statement on the abstract.

Introduction:

  • An unclear gap in the research, relevance, and significance of this study.

Material and Methods

  1. Clear Research area.
  2. The review method is acceptable.

 Results:

  • The result is acceptable and there is progress/improvement in the discussion.

 Conclusion

  • Acceptable conclusions.

References

      Up-dated the old references.

Thank you.

Author Response

My response to Reviewer 1 comments (second round) are as follows:

Title:

The title is acceptable.

No response needed

Abstract:

Unclear of problem statement on abstract

As suggested by this reviewer on the first round of comments, I have added the specific objectives  in the first sentence of the abstract the study. These objectives were extracted from the last paragraph of the Introduction (see track changes) as this reviewer suggested. These two study objectives clearly define the problem statement or goals of the study.  

Introduction

An unclear gap in research, relevance and significance of study. 

The gap in research is clearly defined on line 56 - there is still no historical summary of SEM copper data available based on a variety of land use activities.

To address the relevance concern a phrase was added to the sentence on page 56 (bioavailable potentially toxic form) to demonstrate why SEM copper is relevant. It is relevant because it is the toxic form of copper.

The significance of the study is addressed in lines 53 -57 stating that no historical summary of SEM copper data available based on a variety of land use activities and SEM copper is potentially toxic.

Materials and Methods

  1. Clear research area
  2. The review method is acceptable

No response needed

Results

The result is acceptable and there is progress/improvement in the discussion

No response needed

Conclusions

Acceptable

No response needed

References

Update old references

No response needed

Reviewer 2 Report

In this revised manuscript, the authors still need to consider the following comments.

  1. As for earlier Comment 1, in the second paragraph of the Materials and Methods, the authors just explain the meaning of non-agricultural areas, but what about all study areas, agricultural study areas and reference/control areas?
  2. As for earlier Comment 4: On Page 10, Line 176, it’s better to list and label the equation to calculate the 90th centile on a single line. For example,

The a and b factors of the regression equation were used in the following equation to calculate the 90th centile:

where: probit %=the probit transformed percentage (i.e., if a 90th centile is desired then the probit transformed percentage equal to 90% was used).

  1. As for earlier Comment 11: And in the figures, please pay attention to format of “th”. In “90th”, the “th” should be in superscript. For example, on Line 187, in Table 2, in Figure 5-7.

Author Response

Comments by Reviewer 2 are addressed below:

  1. As for earlier Comment 1, in the second paragraph of the Materials and Methods, the authors just explain the meaning of the non-agricultural areas but what about all study areas, agricultural study areas and reference control areas?

In response to the reviewer all site categories were defined in order: all sites, non-agricultural sites, agricultural sites, and reference/control sites

2. As for comment 4: on page 10 line 176 its better to list and label the equation to calculate the 90th centile on a single line. For example the a and b factors of regression equation were used in the following equation to calculate the 90th centile where probit: % = the probit transformed percentage (i.e., 90th centile is desired then the probit transformed percentage equal to 90% was used).

I don't understand why using a single line is so important. The way the equation is presented is technically correct and I have used this format in other published papers. 

3. As for comment 11:  and in the figures please pay attention to format of "th" in 90th , the "th" should be superscript: For example on line 187 in Table , in Figures 5-7

The "th" can be presented as either on the same line or as a superscript. It really does not make any difference as I have seen it done both ways.  For me to change the 90th centiles in the figures to superscript would require downloading the figures converting back to the original format making the change and then inserting the figures back into the third version of this paper that has already been type set. I don think this change is needed.